# Protective effects of L-arginine on gastric ulcer induced by ethanol in rats: Modulation of oxidative stress

Kaveh Rahimi *, Kosar Hassan Pour, Seyedeh Roghayeh Afandak, Fatemeh Razavi, Fatemeh Shojaat Maharlooei

Department of Basic Sciences, Faculty of Veterinary Medicine, Shahid Chamran University of Ahvaz, Ahvaz, Iran

* k.rahimi@scu.ac.ir, kaveh_rahimi66a@yahoo.com

## Abstract

Gastric ulcers occur due to an imbalance between harmful and protective factors. This study aimed to examine the effect of L-arginine on ethanol-induced gastric ulcers in rats. In this study, 30 male Wistar rats were divided into five groups: a control group that received saline, group 1 which was treated with 70% ethanol to induce gastric ulcers, group 2 that received saline along with L-arginine at a dose of 500 mg/kg administered intraperitoneally, group 3 that was given ethanol along with L-arginine at the same dose but administered orally, and group 4 which received ethanol together with omeprazole at a dose of 10 mg/kg administered intraperitoneally. After the ethanol treatment induced ulceration, gastric lesions were evaluated, and assays were performed to measure antioxidant enzyme levels, malondialdehyde (MDA) levels, and the gene expression of inducible nitric oxide synthase (iNOS) and endothelial nitric oxide synthase (eNOS). Treatment with L-arginine or omeprazole reduced ethanol-induced gastric damage by lowering the ulcer index, increasing gastric pH, improving antioxidant status (decreased MDA, increased SOD and CAT), and modulating NO and iNOS levels. L-arginine showed protective effects comparable to those of omeprazole. This study shows that L-arginine can effectively reduce gastric mucosal injury caused by ethanol, achieving results comparable to those of omeprazole. The protective effects of L-arginine are likely due to its ability to enhance antioxidant defenses, regulate gastric acidity, and modulate nitric oxide pathways. These findings highlight the potential therapeutic role of L-arginine in managing gastric ulcers.

## Introduction

Gastric ulcers are a chronic and recurring condition defined by the repeated formation of ulcers in the mucous membranes of the stomach. These ulcers, which are open sores on the gastric lining, have long been recognized as a significant global

**Data availability statement:** All relevant data are within the paper and its Supporting Information files.

**Funding:** The author(s) received no specific funding for this work.

**Competing interests:** The authors have declared that no competing interests exist.

health issue. Several factors can contribute to the development of gastric ulcers, including infection with Helicobacter pylori, the use of nonsteroidal anti-inflammatory drugs (NSAIDs), and lifestyle choices such as excessive alcohol consumption [1, 2].

Peptic ulcer disease (PUD) remains a significant global health concern despite advances in diagnosis and treatment. In 2019, the global prevalence of PUD was estimated at 8.09 million cases, reflecting a 25.8% increase since 1990, although the age-standardized prevalence rate declined to 99.4 per 100,000 populations. Over the past three decades, disability-adjusted life years (DALYs) attributed to PUD have decreased by more than 60%, yet the burden remains higher among males than females. Regional disparities persist, with South Asia showing the highest age-standardized prevalence and Kiribati ranking first at the national level, while high-income regions demonstrate the lowest mortality rates. Furthermore, a positive association between PUD burden and the sociodemographic index (SDI) highlights the complex interplay between socioeconomic status and disease distribution worldwide [3].

Oxidative stress is considered a primary factor in the development of gastric ulcers [4–7]. Nitric oxide (NO) is a signaling molecule and free radical that plays a key role in regulating gastrointestinal physiology, particularly in the stomach. Under normal conditions, NO contributes to gastric protection by increasing mucosal blood flow, inhibiting acid secretion, and supporting defense mechanisms [8]. In this state, endothelial nitric oxide synthase (eNOS) expressed in endothelial cells produces low and tightly regulated levels of NO with protective effects. In contrast, inducible nitric oxide synthase (iNOS) is activated in response to inflammation or harmful stimuli such as alcohol, generating large amounts of NO. Excess NO can react with reactive oxygen species to form compounds such as peroxynitrite ($ONOO^-$), which can damage proteins, lipids, and DNA [9]. NO has been recognized for various biological functions, including its role in wound healing. NO is synthesized by nitric oxide synthase from the amino acid L-arginine. Studies have shown that insufficient dietary L-arginine can slow down wound healing in experimental settings. Additionally, L-arginine can be processed by the enzyme arginase-1 into urea and ornithine, leading to the production of L-proline—essential for collagen formation—and polyamines, which promote cell growth [10].

Since comprehensive research on the protective effects of L-arginine against ethanol-induced gastric ulcers and its role in regulating oxidative stress in animal models has not been conducted, the primary aim of this study was to investigate the potential effects of L-arginine in preventing and improving ethanol-induced gastrointestinal damage in the gastric tissue of rats.

## Methods

### Animals

In this study, 30 adult male Wistar rats weighing between 200–250 g were used. All animals were housed under standard laboratory conditions, maintained at a temperature of $22 \pm 2°C$, with a 12-hour light/dark cycle, and provided ad libitum access to both food and water.

All experimental procedures were approved by the Ethical Committee of the Faculty of Veterinary Medicine at Shahid Chamran University of Ahvaz (Approval No. IR.SCU.REC.1404.097) and were performed in accordance with national guidelines for the care and use of laboratory animals, the ARRIVE guidelines [11], and the AVMA recommendations for euthanasia [12]. Rats were anesthetized via intraperitoneal injection of ketamine (50 mg/kg) combined with xylazine (4 mg/kg), and depth of anesthesia was confirmed by loss of the pedal withdrawal reflex before any procedure. At the end of the experimental period, animals were humanely euthanized under deep anesthesia using an overdose of ketamine–xylazine, followed by cervical dislocation, in compliance with AVMA guidelines. Throughout the 14-day study period, animals were monitored twice daily for clinical signs, behavior, and overall health. Humane endpoints were applied to minimize pain and distress. All procedures were conducted by trained personnel, and no unexpected deaths occurred.

### Experimental design

The study was conducted as a randomized experimental trial. Rats were randomly divided into five groups as follows: Control Group: Rats received saline solution. Gastric Ulcer Group (Eth): Rats were administered ethanol (Merck, USA). L-Arginine Group (L-ARG 500): Rats received saline, followed by L-arginine (Sigma, USA) at a dose of 500 mg/kg body weight [13] intraperitonically for 14 days. L-Arginine + Ethanol Group (Eth + L-ARG 500): Rats received ethanol, followed by L-arginine (500 mg/kg, p.o.) for 14 days. Omeprazole Group (Eth + L-OMP 500): Rats received ethanol, followed by omeprazole (20 mg/kg, i.p., [14] Alborz Darou Pharmaceutical Co. Iran) for 14 days.

### Induction of gastric ulcer

Animals underwent a 24-hour fasting period with free access to water. Gastric ulcers were induced by oral administration of 96% ethanol at a dose of 1 mL/200 g body weight via gastric gavage. Rats were euthanized 90 minutes after ethanol administration to collect gastric tissues [14–16].

### Evaluation of gastric lesions

After treatment, the stomachs were excised and opened along the greater curvature. The gastric mucosa was rinsed with saline, and lesions were assessed macroscopically. The ulcer index was calculated according to reference [15]. The percentage of ulcer inhibition was calculated using the following formula:

Inhibition (%) = [(Ulcer Index of Ulcer Group – Ulcer Index of Treated Group) ÷ Ulcer Index of Ulcer Group] × 100 where UI is the ulcer index.

### Gastric juice pH

Gastric juice was collected from the stomach after euthanasia, centrifuged at 3000 rpm for 10 minutes, and the supernatant pH was measured using a calibrated digital pH meter (Metrohm, Switzerland).

### Nitric Oxide (NO) assay

NO levels in gastric tissue homogenates were determined using the Griess reaction kit (ZellBio GmbH, Germany). Briefly, nitrite concentration, a stable NO metabolite, was quantified colorimetrically at 540 nm and expressed as μmol/g tissue.

### Oxidative stress status in gastric tissue

The oxidative stress status in gastric tissue was evaluated using commercial assay kits (ZellBio GmbH, Germany). Malondialdehyde (MDA) levels were determined by the thiobarbituric acid reactive substances (TBARS) method, and absorbance was recorded at 532 nm, with results expressed as nmol/mg protein. Superoxide dismutase (SOD) activity was measured based on the inhibition of nitroblue tetrazolium (NBT) reduction at 560 nm, and values were expressed

as U/mg protein. Catalase (CAT) activity was assessed by monitoring the decomposition of hydrogen peroxide at 240 nm, with results expressed as U/mg protein. Glutathione (GSH) levels were determined using the DTNB method at 412 nm, and results were expressed as µmol/g tissue. In all assays, protein concentration was quantified using the Bradford method for normalization.

### Gene expression analysis (RT-qPCR)

The expression of inducible nitric oxide synthase (iNOS or Nos2) and endothelial nitric oxide synthase (eNOS or Nos3) genes was evaluated by quantitative real-time RT-PCR. Total RNA was extracted from gastric tissues using an RNA extraction kit (Qiagen RNeasy Mini Kit, Germany). cDNA was synthesized from 1 µg RNA using a reverse transcription kit (Thermo Fisher Scientific, USA). Real-time PCR was performed using SYBR Green PCR Master Mix (Applied Biosystems, USA) on a StepOnePlus Real-Time PCR System (Applied Biosystems, USA). Primer sequences were as follows: iNOS — forward 5′-CAC GAC ACC CTT CAC CAC AAG-3′, reverse 5′-TTG AGG CAG AAG CTC CTC CA-3′; eNOS— forward 5′-TTC CGG CTG CCA CCT GAT CCT AA-3′, reverse 5′-AAC ATA TGT CCT TGC TCA AGG CA-3′; Gapdh (housekeeping gene) — forward 5′-TGC ACC ACC AAC TGC TTA GC-3′, reverse 5′-GGC ATG GAC TGT GGT CAT GAG-3′. PCR conditions were as follows: initial denaturation at 95°C for 5 min, followed by 35 cycles of denaturation at 95°C for 30 s, annealing at 55–60°C for 30 s, and extension at 72°C for 1 min, with a final extension at 72°C for 10 min. Melt-curve analysis was performed to confirm amplification specificity. Relative gene expression was calculated using the $2^{-\Delta\Delta Ct}$ method with Gapdh as the internal control, and results were expressed as fold-change relative to the control group.

### Statistical analysis

The data were analyzed using GraphPad Prism 8 software. The normality of the data was initially assessed using the Shapiro-Wilk test. One-way analysis of variance (ANOVA) was used to compare the means of different groups. In case of significant differences, Tukey's post hoc test was applied for pairwise comparisons. For non-parametric analyses, the Kruskal-Wallis test was used. A significance level of $p \leq 0.05$ was considered statistically significant Fig 1.

## Results

### Ulcer index

Administration of ethanol markedly increased the ulcer index ($14.83 \pm 0.87$) compared with the control group ($0.00 \pm 0.00$, $p < 0.0001$). Pretreatment with L-arginine (500 mg/kg) completely prevented ulcer formation, with an index comparable to controls ($0.00 \pm 0.00$, $p > 0.9999$). Both the Eth + L-ARG and Eth + OMP groups showed significant reductions in ulcer index ($5.83 \pm 0.31$ and $7.50 \pm 0.43$, respectively; $p < 0.0001$ vs. Eth), although their values remained higher than the L-ARG group ($p < 0.0001$). No significant difference was detected between the Eth + L-ARG and Eth + OMP groups ($p = 0.1038$) (Figs 2 and 3).

### Percentage of ulcer inhibition

Consistent with the ulcer index data, ethanol completely abolished ulcer inhibition ($0.00 \pm 0.00$), while L-arginine alone preserved full protection ($100.0 \pm 0.00$). Co-treatment with L-arginine or omeprazole significantly increased ulcer inhibition ($59.99 \pm 3.06$ and $49.15 \pm 2.49$, respectively; both $p < 0.0001$ vs. Eth). However, their effects remained inferior to L-arginine alone ($p < 0.0001$). Notably, inhibition was slightly but significantly higher in Eth + L-ARG compared with Eth + OMP ($p = 0.0017$) (Figs 3 and 4).

### Gastric pH

Ethanol markedly decreased gastric pH ($2.52 \pm 0.09$) relative to controls ($3.68 \pm 0.11$, $p < 0.0001$). L-arginine alone maintained pH within the control range ($3.67 \pm 0.16$, $p > 0.9999$). Both L-arginine and omeprazole co-treatment significantly

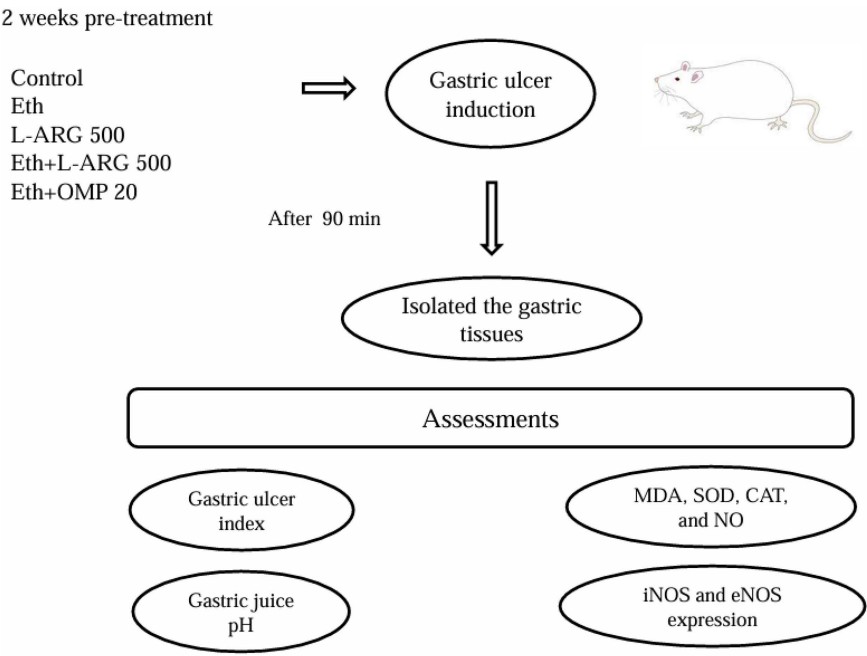

2 weeks pre-treatment

Control
Eth
L-ARG 500
Eth+L-ARG 500
Eth+OMP 20

Gastric ulcer
induction

After 90 min

Isolated the gastric
tissues

Assessments

Gastric ulcer
index

MDA, SOD, CAT,
and NO

Gastric juice
pH

iNOS and eNOS
expression

**Fig 1. Graphical abstract summarizing the protective effects of L-arginine against ethanol-induced gastric ulcer in rats.**

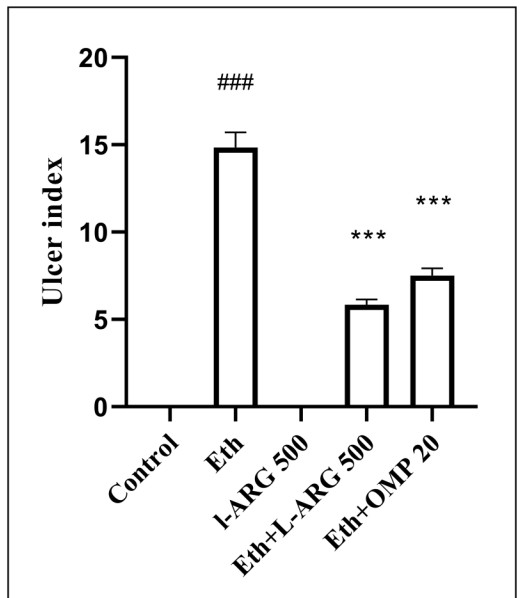

**Fig 2. Effects of L-ARG on ulcer index.** Data are presented as Mean ± SEM (n = 6). Groups: Control, Ethanol (Eth), L-ARG 500 mg/kg, L-ARG 500 mg/kg + Eth, and Omeprazole 20 mg/kg + Eth. #p < 0.05, ##p < 0.01, ###p < 0.001 vs. control; *p < 0.05, **p < 0.01, ***p < 0.001 vs. Eth.

attenuated the ethanol-induced decrease (3.40 ± 0.17 and 3.36 ± 0.13, respectively; p < 0.01 vs. Eth), with no differences among treated groups (all p > 0.5) (Fig 5).

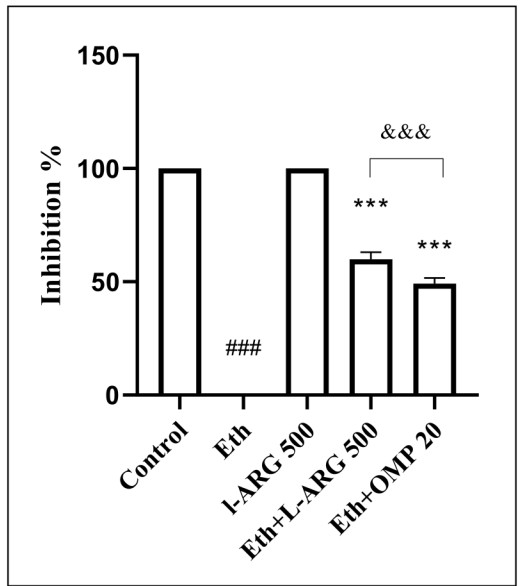

**Fig 3. Effects of L-ARG on % inhibition of ulcer.** Data are presented as Mean ± SEM (n = 6). Same groups and significance as in 2A. Groups: Control, Ethanol (Eth), L-ARG 500 mg/kg, L-ARG 500 mg/kg + Eth, and Omeprazole 20 mg/kg + Eth. #p < 0.05, ##p < 0.01, ###p < 0.001 vs. control; *p < 0.05, **p < 0.01, ***p < 0.001 vs. Eth.

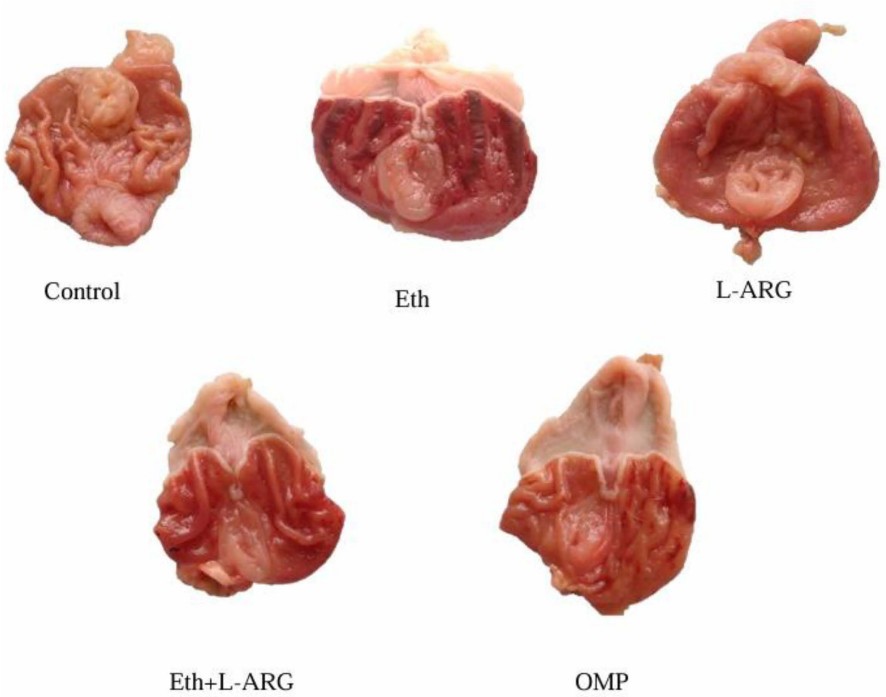

**Fig 4. Representative macroscopic photographs of rat stomachs from each experimental group.**

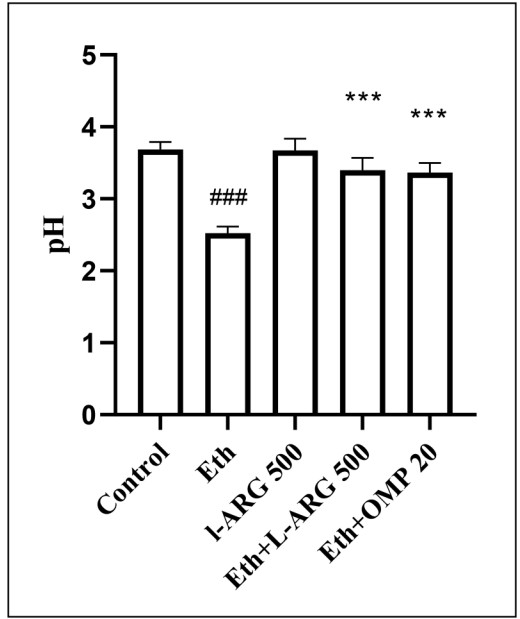

**Fig 5. Effects of L-ARG on gastric pH.** Data are presented as Mean ±SEM (n = 6). Control, Ethanol (Eth), L-ARG 500 mg/kg, L-ARG 500 mg/kg + Eth, and Omeprazole 20 mg/kg + Eth groups. #p < 0.05, ##p < 0.01, and ###p < 0.001 show a significant level difference compared to the control group. *p < 0.05, **p < 0.01, and ***p < 0.001 show a significant level difference compared to the Eth group.

## Oxidative stress parameters (MDA, SOD, CAT, NO)

Ethanol exposure significantly increased lipid peroxidation, as indicated by elevated MDA levels ($4.31 \pm 0.22$ vs. $1.96 \pm 0.12$ in controls, $p < 0.0001$). L-arginine alone normalized MDA, whereas both Eth + L-ARG and Eth + OMP groups showed partial but significant reductions compared with Eth ($\approx 3.0$, $p < 0.0001$), with no difference between them ($p > 0.9999$) (Fig 6).

Antioxidant defenses were strongly impaired by ethanol, as shown by reduced SOD ($30.74 \pm 1.79$ vs. $70.00 \pm 0.76$ in controls, $p < 0.0001$) and CAT activities ($20.13 \pm 1.10$ vs. $35.90 \pm 1.34$, $p < 0.0001$). L-arginine restored both enzymes close to control levels. Eth + L-ARG and Eth + OMP also improved SOD ($56.97 \pm 1.63$ and $50.22 \pm 1.55$, respectively) and CAT ($25.82 \pm 0.63$ and $25.46 \pm 0.44$, respectively) compared with Eth, although values remained lower than L-ARG alone (Figs 7 and 8).

Similarly, nitric oxide (NO) production was markedly reduced in ethanol-treated rats ($110.9 \pm 3.85$ vs. $181.0 \pm 3.47$ in controls, $p < 0.0001$). L-arginine preserved NO levels, while both Eth + L-ARG and Eth + OMP significantly improved NO compared with Eth ($154.9 \pm 2.56$ and $148.6 \pm 2.57$, respectively; $p < 0.0001$), without difference between them ($p = 0.5901$) (Fig 9).

## Gene expression (*iNOS* and *eNOS*)

Ethanol markedly upregulated *iNOS* expression ($3.81 \pm 0.23$ vs. $1.01 \pm 0.01$ in controls, $p < 0.0001$), while L-arginine maintained basal levels ($0.995 \pm 0.01$, $p > 0.9999$ vs. control). Both Eth + L-ARG and Eth + OMP significantly suppressed ethanol-induced *iNOS* overexpression ($2.48 \pm 0.20$ and $2.51 \pm 0.16$, $p < 0.0001$ vs. Eth), with no differences between them ($p > 0.9999$) (Fig 10).

Conversely, *eNOS* expression was downregulated by ethanol ($0.485 \pm 0.046$ vs. $1.007 \pm 0.011$ in controls, $p < 0.0001$). L-arginine alone preserved normal expression. Both Eth + L-ARG and Eth + OMP significantly enhanced *eNOS* compared with Eth ($1.238 \pm 0.017$ and $1.202 \pm 0.032$, $p < 0.0001$), and values were even higher than L-ARG alone ($p < 0.001$). No difference was observed between Eth + L-ARG and Eth + OMP ($p = 0.8664$) (Fig 11).

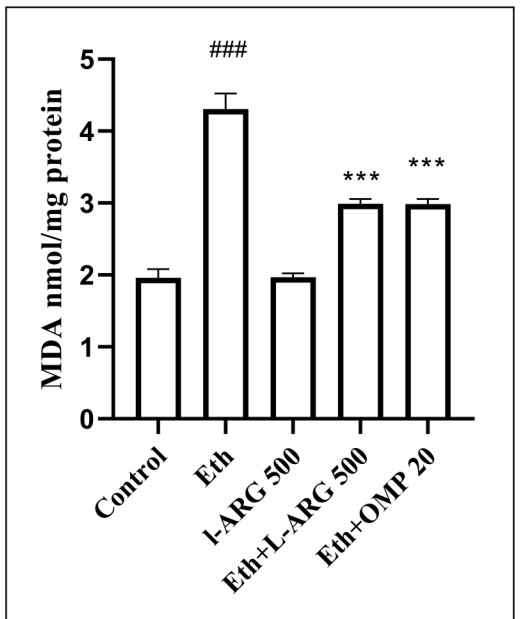

**Fig 6. Effects of L-ARG on MDA levels, indicating lipid peroxidation.** Data are presented as Mean±SEM (n=6). Groups and significance as in previous figures. Groups: Control, Ethanol (Eth), L-ARG 500 mg/kg, L-ARG 500 mg/kg+Eth, and Omeprazole 20 mg/kg+Eth. #p<0.05, ##p<0.01, ###p<0.001 vs. control; *p<0.05, **p<0.01, ***p<0.001 vs. Eth.

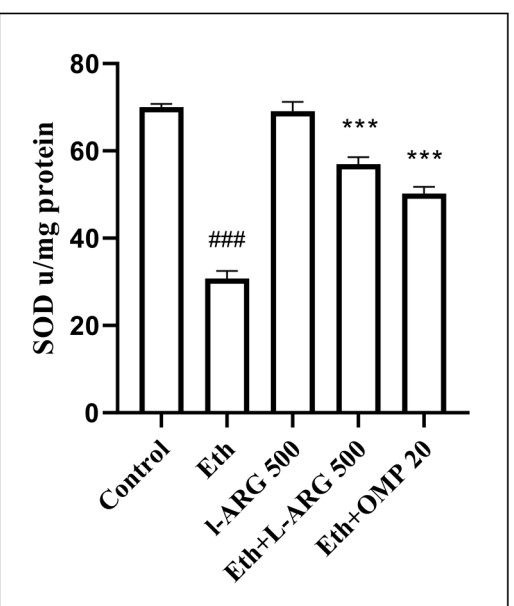

**Fig 7. Effects of L-ARG on SOD activity, showing antioxidant defense.** Data are presented as Mean±SEM (n=6). Groups: Control, Ethanol (Eth), L-ARG 500 mg/kg, L-ARG 500 mg/kg+Eth, and Omeprazole 20 mg/kg+Eth. #p<0.05, ##p<0.01, ###p<0.001 vs. control; *p<0.05, **p<0.01, ***p<0.001 vs. Eth.

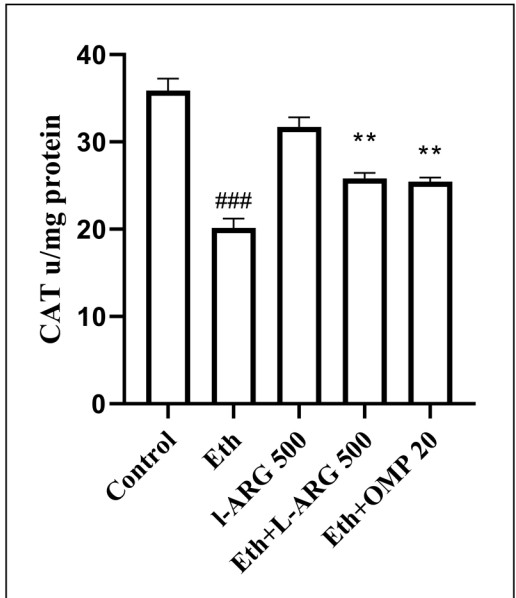

**Fig 8. Effects of L-ARG on CAT activity, showing antioxidant defense.** Data are presented as Mean±SEM (n=6). Groups: Control, Ethanol (Eth), L-ARG 500 mg/kg, L-ARG 500 mg/kg+Eth, and Omeprazole 20 mg/kg+Eth. #p<0.05, ##p<0.01, ###p<0.001 vs. control; *p<0.05, **p<0.01, ***p<0.001 vs. Eth.

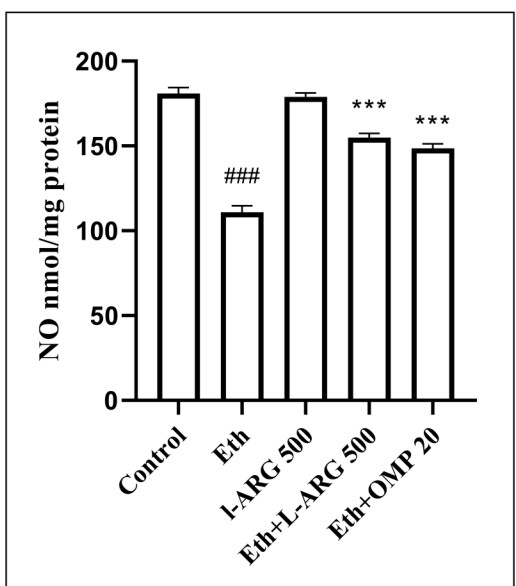

**Fig 9. Effects of L-ARG on nitric oxide (NO) levels.** Data are presented as Mean±SEM (n=6). Groups: Control, Ethanol (Eth), L-ARG 500 mg/kg, L-ARG 500 mg/kg+Eth, and Omeprazole 20 mg/kg+Eth. #p<0.05, ##p<0.01, ###p<0.001 vs. control; *p<0.05, **p<0.01, ***p<0.001 vs. Eth.

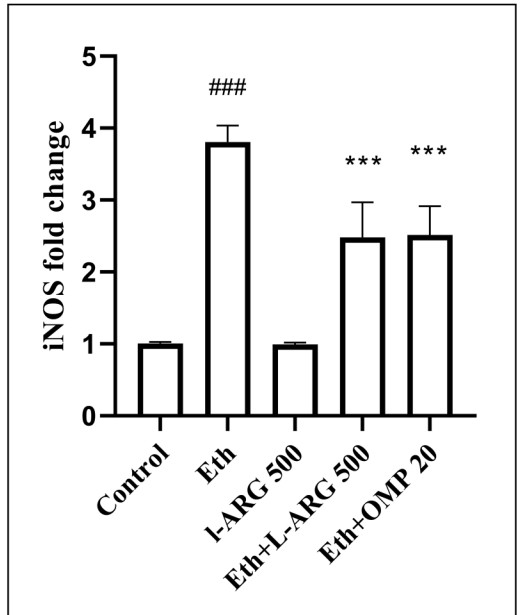

**Fig 10. Effects of L-ARG on iNOS gene expression.** Data are presented as Mean ± SEM (n = 6). Groups and significance as described previously. Groups: Control, Ethanol (Eth), L-ARG 500 mg/kg, L-ARG 500 mg/kg + Eth, and Omeprazole 20 mg/kg + Eth. #p < 0.05, ##p < 0.01, ###p < 0.001 vs. control; *p < 0.05, **p < 0.01, ***p < 0.001 vs. Eth.

## Discussion

Ethanol disrupts the balance between protective and harmful factors in the stomach, damaging the gastric mucosa by impairing blood flow and increasing oxidative stress. This leads to elevated reactive oxygen species and mitochondrial dysfunction, which are strongly linked to gastric injury. As ethanol concentration and exposure duration increase, so do lipid peroxidation and mitochondrial damage [17]. In a physiological study using an *ex vivo* rat gastric chamber model, researchers found that ethanol can nitrosylate endogenously produced NO. When saline containing ethanol was infused into the stomach, the levels of nitrite and nitrate in the gastric luminal fluid decreased by approximately 70%. However, these levels returned to baseline after the infusion was replaced with an ethanol-free solution. At the same time, the levels of NO in the gastric mucosa increased slightly, indicating that ethanol does not inhibit the synthesis of NO. These findings suggest that ethanol may absorb NO originating from the gastric tissue and could potentially react with NO-derived species, such as $N_2O_3$, to form a compound called ethyl nitrite. This decrease in NO can stem from either impaired eNOS activity or dysregulated iNOS expression, both influenced by oxidative stress and inflammatory conditions [18]. In our study dministration of Eth significantly increased the ulcer index and reduced gastric pH compared to the control group. It also led to a marked decrease in antioxidant enzyme levels, including SOD and CAT, as well as a reduction in the percentage of ulcer inhibition. Ethanol exposure elevated levels of MDA and inducible iNOS, while concurrently decreasing the levels of SOD, CAT, and NO. These biochemical changes reflect the oxidative stress and inflammatory damage induced by ethanol, which disrupts the antioxidant defense system and alters nitric oxide signaling pathways, thereby exacerbating gastric mucosal injury.

L-arginine is an essential amino acid involved in numerous physiological functions within the human body [19–23]. It serves as a key precursor for the production of nitric oxide (NO), a powerful vasodilator and neurotransmitter [19]. Besides its role in NO synthesis, L-arginine contributes to protein synthesis, supports immune system function, and plays an important role in wound healing processes [24]. This amino acid is mainly supplied through dietary intake from sources such as meat, poultry, fish, dairy products, and nuts [19], although it can also be endogenously produced from other amino acids like glutamate and

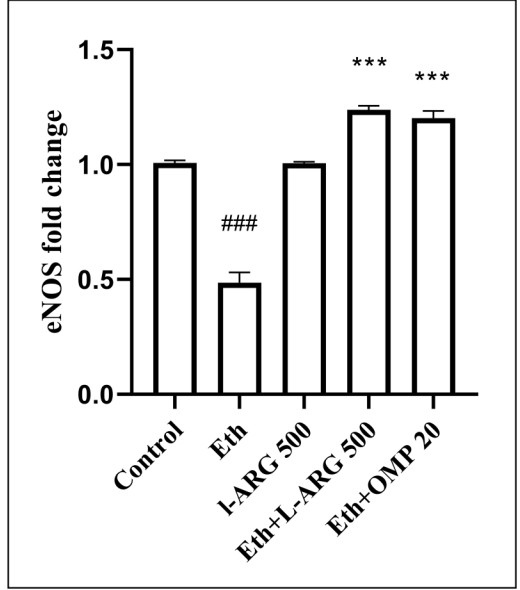

**Fig 11. Effects of L-ARG on eNOS gene expression.** Data are presented as Mean ± SEM (n = 6). Groups: Control, Ethanol (Eth), L-ARG 500 mg/kg, L-ARG 500 mg/kg + Eth, and Omeprazole 20 mg/kg + Eth. #p < 0.05, ##p < 0.01, ###p < 0.001 vs. control; *p < 0.05, **p < 0.01, ***p < 0.001 vs. Eth.

glutamine. Oxidative stress, characterized by an imbalance between free radical generation and the body's antioxidant capacity, significantly contributes to the development of various diseases including cardiovascular disorders, neurodegenerative conditions, and cancer [25, 26]. Consequently, there is a continuous demand for L-arginine capable of mitigating oxidative stress and helping to prevent these health issues [27–30]. In a study, the administration of L-arginine reversed the negative effects of resveratrol in rats with gastric ulcers induced by indomethacin. This effect was linked to an increase in the expression of eNOS and a decrease in the expression of iNOS in gastric tissue. Additionally, the administration of L-arginine reduced myeloperoxidase activity and accelerated the healing of ulcers [31]. In the current study, the co-administration of L-arginine (500 mg/kg) or omeprazole (20 mg/kg) significantly reduced the ulcer index and increased the percentage of ulcer inhibition compared to the ethanol group. Treatment with L-arginine effectively reversed the ethanol-induced decrease in antioxidant enzyme levels and NO levels while also lowering MDA and iNOS expression. These findings suggest that L-arginine has strong antioxidant and protective properties. Additionally, eNOS levels were maintained, and iNOS expression was downregulated in the treated groups. Importantly, L-arginine alone did not cause significant changes compared to the control group, indicating its safety under normal physiological conditions. Overall, L-arginine alleviates ethanol-induced gastric damage through its antioxidant activity, regulation of gastric acidity, and modulation of nitric oxide pathways.

This study provides novel evidence on the gastroprotective potential of L-arginine using validated macroscopic and biochemical endpoints. Nevertheless, some limitations should be considered. Only a single dose of L-arginine was evaluated, which precluded dose–response analysis. Histopathological confirmation and assessment of downstream mediators such as inflammatory cytokines or apoptosis markers were not performed, limiting mechanistic depth. In addition, the use of only male rats helped reduce hormonal variability but may restrict the generalizability of the findings. Future investigations addressing these points may provide a more comprehensive understanding of L-arginine's effects.

## Conclusion

The findings of this study demonstrated that L-arginine has a significant protective effect against ethanol-induced gastric ulcers, likely through improving oxidative status, increasing antioxidant enzyme activity, and regulating the balance

between iNOS and eNOS. These results are consistent with previous studies reporting the protective role of L-arginine in the gastrointestinal system. However, the use of a single drug dose and the lack of histological examinations are among the limitations of the present study. Future research can achieve a more precise understanding of the involved molecular mechanisms by expanding the experimental design and employing various ulcer models.

## Supporting information

**S1 File. Raw data file (Excel format) containing all original measurements collected in the study, including participant/sample identifiers and recorded variables.**
(XLSX)

## Acknowledgments

We are grateful to the University of Ahvaz, Ahvaz, Iran (Number: SCU.VB1404.50857).

## Author contributions

**Data curation:** kaveh rahimi.

**Investigation:** kaveh rahimi, Kosar Hassan Pour, Seyedeh Roghayeh Afandak, Fatemeh Razavi, Fatemeh Shojaat Maharlooei.

**Methodology:** kaveh rahimi.

**Project administration:** kaveh rahimi.

**Software:** kaveh rahimi.

**Supervision:** kaveh rahimi.

**Writing – original draft:** kaveh rahimi, Kosar Hassan Pour, Seyedeh Roghayeh Afandak, Fatemeh Razavi, Fatemeh Shojaat Maharlooei.

**Writing – review & editing:** kaveh rahimi, Kosar Hassan Pour, Seyedeh Roghayeh Afandak, Fatemeh Razavi, Fatemeh Shojaat Maharlooei.

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
