## [Decision Letter · Decision Letter 0]

29 Sep 2025

Dear Dr. rahimi,

We look forward to receiving your revised manuscript.

Kind regards,

Partha Mukhopadhyay, Ph.D.

Section Editor

PLOS ONE

Journal Requirements:

2. To comply with PLOS One submissions requirements, in your Methods section, please provide additional information regarding the experiments involving animals and ensure you have included details on (1) methods of sacrifice, (2) methods of anesthesia and/or analgesia, and (3) efforts to alleviate suffering.

[We are grateful to the Research Council of Shahid Chamran University of Ahvaz for financial support (GN: SCU.VB1404.50857).]

[The author(s) received no specific funding for this work.]

6. Please include a separate caption for each figure in your manuscript.

Reviewers' comments:

Reviewer's Responses to Questions

**Comments to the Author**

1. Is the manuscript technically sound, and do the data support the conclusions?

Reviewer #1: Yes

Reviewer #2: Partly

2. Has the statistical analysis been performed appropriately and rigorously?

Reviewer #1: Yes

Reviewer #2: Yes

3. Have the authors made all data underlying the findings in their manuscript fully available?

Reviewer #1: Yes

Reviewer #2: No

4. Is the manuscript presented in an intelligible fashion and written in standard English?

Reviewer #1: No

Reviewer #2: Yes

Reviewer #1: The authors evaluate the use of L-Arginine on alcohol-induced gastric ulcers. They observed that prior treatment with per os (p.o.) L-arginine for 14 days is protective against the development of gastric ulcers, mediated by oxidative stress and nitric oxide. The manuscript is sound and technically well-written, although there are some typos and sections that need improvement, which I will detail in my review, divided in topics.

Format:

Please number the lines of the text to make it easier to reference specific points in the review. Also, standardize the references by adding DOI and PMCID whenever possible.

Terminology:

Some terms need to be written in italics, such as species names (e.g., Helicobacter pylori), Latin terms (e.g., et al., in vivo, ex vivo, ad libitum), and gene names.

Introduction:

This section lacks epidemiological information on alcohol-induced ulcers in human patients and references related to alcohol-induced gastric ulcers in general. Additionally, some paragraphs contain long sequences of sentences and statements without references. Please double-check references 3 and 6, as they do not seem to support their respective preceding statements.

Materials and Methods:

This section is quite messy and needs to be well-described for reproducibility. When describing the animal study, include the ethical statement information here, not at the end of the manuscript. You can repeat the ethics approval number at the end if needed, but it is important to include this information in the methods section.

Experimental design:

This subsection is confusing as it is currently described with minimal context. It would be better to describe the animal experiments and induction of gastric ulcers before detailing the experimental design. Also, introduce the names of the groups as they will be referred to throughout the manuscript and in the figures to make the text more comprehensible. Make sure to clearly specify your protocol. It took me some time to understand that the induction of gastric ulcers was a single dose administered a few minutes before euthanizing the specimens and collecting the stomachs. It is not even clear how long after gavaging with ethanol the animals were euthanized.

Was the induction of gastric ulcers done with 1 mL/100 g of body weight? If so, why was this different from other publications cited by the authors, where 1 mL/200 g was used?

You don't mention the manufacturer of the omeprazole. Also, omeprazole takes advantage of prior activation through an acidic environment before absorption. Why did you choose intraperitoneal administration of omeprazole instead of p.o.?

Evaluation of Gastric Lesions:

You can be more descriptive and refer to reference 12 as a source for the equations. The authors also don't mention the calculation of ulcer inhibition, which is presented in the results.

Oxidative Stress Status in Gastric Tissue:

In this section, specify the manufacturer of the kits used for each assay. Additionally, I miss the results of the glutathione levels mentioned here but not shown in the results.

Real-Time RT-PCR:

It is important to note that the authors are measuring mRNA, which reflects gene expression, not necessarily protein level. Therefore, the correct way to refer to the targets is by their gene names, always in italics, and in rodent models, with the first letter capitalized and the subsequent letters in lowercase. Please inform which kits were used for RNA extraction, reverse transcription, and PCR itself (Taqman? SYBR Green?). Also, the sentence immediately after the eNOS sequence (which should be named Nos3) contains an extra GAPDH that doesn't make sense.

Experimental methods missing:

Two results are missing from the description in the materials and methods section: assessment of gastric juice pH and nitric oxide levels measurement.

Results Section:

The results section is extremely unappealing for the reader. It is monotonous and repetitive. It sounds like a long copy-paste. The description in the last paragraph of the section "MDA, SOD, CAT, and NO levels" doesn't match the graphics in figure 4D. Please double-check it.

The last paragraph of the results is evidence of the copy-paste issue mentioned earlier. It describes the results of SOD, but I suppose it should be Nos2 gene expression.

Nevertheless, the results section should be rewritten in a more engaging format, with due attention to the details.

Discussion:

In the first paragraph, the term (TBARS) doesn't make sense in the context of the preceding words. The following sentence mentions a study that is not referenced here. I suppose it is reference 14.

Finally, fix the typos in the "Credit authorship..." section.

I appreciate the authors for acknowledging the limitations of their study. Indeed, simple histological stainings are not very expensive and would be valuable for demonstrating microscopic alterations. Additionally, immunostaining to highlight immune cells could significantly enhance your data. Please consider incorporating these methods into future projects involving gastric ulcers.

Reviewer #2: Summary

Rahimi et al., investigate the protective effects of L-arginine against ethanol-induced gastric ulcers in a rat model. The authors assess parameters like gastric ulcer index, gastric pH, oxidative stress markers (MDA, SOD, CAT, GSH), and iNOS/eNOS levels, with the data indicating that L-arginine reduces ethanol-induced gastric injury similarly to omeprazole, primarily by modulating oxidative stress and nitric oxide pathways. This paper topic explores the potential therapeutic role of a widely available amino acid in ulcer management. However, several major concerns regarding study design, methodology, and reporting need to be addressed before the manuscript can be considered for publication.

Major comments

1. The study employs only a single dose of L-arginine (500 mg/kg) and omeprazole (20 mg/kg), without justification. A rationale for dose selection should be provided, and the absence of a dose–response analysis limits interpretability and translational relevance.

2. Ulcer evaluation is based solely on macroscopic scoring and biochemical assays. Histological examination to confirm mucosal protection will reinforce the author’s conclusions and validate mechanistic claims.

3. Along the same lines, while oxidative stress and NO pathways are measured, no downstream signaling data measuring inflammatory cytokines, apoptosis markers are assessed. This limits mechanistic insight.

4. Omeprazole is used as a comparator in this study. But it primarily acts as a proton pump inhibitor and not as an antioxidant. Including an antioxidant reference compound would provide a more direct mechanistic comparison.

5. Raw data should be made available as part of the Supplement.

Minor comments

1. Abbreviations such as “Eth,” “OMP,” and “L-ARG 500” should be defined consistently at first mention in both text and figure legends.

2. Exact p-values are not reported, instead, thresholds (p < 0.05, p < 0.01) are given. Reporting actual p-values would improve transparency.

3. Is there a reason why the authors only used male rats in this study? This limits generalizability, as gastric ulcer susceptibility and healing can differ by sex. This limitation should be explicitly acknowledged in the Discussion section.

**Do you want your identity to be public for this peer review?** For information about this choice, including consent withdrawal, please see our Privacy Policy

Reviewer #1: No

Reviewer #2: No

---

## [Author Response · Author response to Decision Letter 1]

6 Oct 2025

Dear Editor,

We sincerely thank you for your constructive comments and guidance regarding our manuscript. We carefully revised the manuscript in line with the editorial requests:

• We have provided additional details in the Methods section regarding methods of anesthesia, euthanasia, and efforts to alleviate suffering.

• We have removed all funding-related text from the Acknowledgments and updated the Funding Statement accordingly.

• The ethics statement now appears only in the Methods section.

• We have clarified our Data Availability Statement and planned for data sharing in accordance with PLOS ONE policy.

• Separate captions have been included for each figure.

We appreciate your support in improving the clarity and compliance of our work. In the following document, we provide a point-by-point response to the reviewers’ comments, indicating the revisions made in the manuscript.

Sincerely

Review Comments to the Author

Reviewer #1: The authors evaluate the use of L-Arginine on alcohol-induced gastric ulcers. They observed that prior treatment with per os (p.o.) L-arginine for 14 days is protective against the development of gastric ulcers, mediated by oxidative stress and nitric oxide. The manuscript is sound and technically well-written, although there are some typos and sections that need improvement, which I will detail in my review, divided in topics.

Format:

Please number the lines of the text to make it easier to reference specific points in the review. Also, standardize the references by adding DOI and PMCID whenever possible. The corrections were made.

Terminology:

Some terms need to be written in italics, such as species names (e.g., Helicobacter pylori), Latin terms (e.g., et al., in vivo, ex vivo, ad libitum), and gene names. The corrections were made.

Introduction:

This section lacks epidemiological information on alcohol-induced ulcers in human patients and references related to alcohol-induced gastric ulcers in general. Additionally, some paragraphs contain long sequences of sentences and statements without references. Please double-check references 3 and 6, as they do not seem to support their respective preceding statements. The corrections were made.

Materials and Methods:

This section is quite messy and needs to be well-described for reproducibility. When describing the animal study, include the ethical statement information here, not at the end of the manuscript. You can repeat the ethics approval number at the end if needed, but it is important to include this information in the methods section.

Experimental design:

This subsection is confusing as it is currently described with minimal context. It would be better to describe the animal experiments and induction of gastric ulcers before detailing the experimental design. Also, introduce the names of the groups as they will be referred to throughout the manuscript and in the figures to make the text more comprehensible. Make sure to clearly specify your protocol. It took me some time to understand that the induction of gastric ulcers was a single dose administered a few minutes before euthanizing the specimens and collecting the stomachs. It is not even clear how long after gavaging with ethanol the animals were euthanized.

Was the induction of gastric ulcers done with 1 mL/100 g of body weight? If so, why was this different from other publications cited by the authors, where 1 mL/200 g was used?

You don't mention the manufacturer of the omeprazole. Also, omeprazole takes advantage of prior activation through an acidic environment before absorption. Why did you choose intraperitoneal administration of omeprazole instead of p.o.?

Evaluation of Gastric Lesions:

You can be more descriptive and refer to reference 12 as a source for the equations. The authors also don't mention the calculation of ulcer inhibition, which is presented in the results.

Oxidative Stress Status in Gastric Tissue:

In this section, specify the manufacturer of the kits used for each assay. Additionally, I miss the results of the glutathione levels mentioned here but not shown in the results.

Real-Time RT-PCR:

It is important to note that the authors are measuring mRNA, which reflects gene expression, not necessarily protein level. Therefore, the correct way to refer to the targets is by their gene names, always in italics, and in rodent models, with the first letter capitalized and the subsequent letters in lowercase. Please inform which kits were used for RNA extraction, reverse transcription, and PCR itself (Taqman? SYBR Green?). Also, the sentence immediately after the eNOS sequence (which should be named Nos3) contains an extra GAPDH that doesn't make sense.

Experimental methods missing:

Two results are missing from the description in the materials and methods section: assessment of gastric juice pH and nitric oxide levels measurement.

All comments have been carefully addressed, and the revised Materials and Methods section is now more detailed, structured, and reproducible. We greatly appreciate your constructive feedback, which has significantly improved the quality of our manuscript.

Results Section:

The results section is extremely unappealing for the reader. It is monotonous and repetitive. It sounds like a long copy-paste. The description in the last paragraph of the section "MDA, SOD, CAT, and NO levels" doesn't match the graphics in figure 4D. Please double-check it.

The last paragraph of the results is evidence of the copy-paste issue mentioned earlier. It describes the results of SOD, but I suppose it should be Nos2 gene expression.

Nevertheless, the results section should be rewritten in a more engaging format, with due attention to the details. The Results section has been completely rewritten. All mean ± SEM values and exact p-values are now clearly reported, and the narrative style has been improved to ensure accuracy, clarity, and readability. The revised version avoids repetition, follows the data shown in the figures, and presents the findings in a more engaging and scientifically rigorous manner.

Discussion:

In the first paragraph, the term (TBARS) doesn't make sense in the context of the preceding words. The following sentence mentions a study that is not referenced here. I suppose it is reference 14.

Finally, fix the typos in the "Credit authorship..." section.

I appreciate the authors for acknowledging the limitations of their study. Indeed, simple histological stainings are not very expensive and would be valuable for demonstrating microscopic alterations. Additionally, immunostaining to highlight immune cells could significantly enhance your data. Please consider incorporating these methods into future projects involving gastric ulcers. The corrections were made.

Reviewer #2: Summary

Rahimi et al., investigate the protective effects of L-arginine against ethanol-induced gastric ulcers in a rat model. The authors assess parameters like gastric ulcer index, gastric pH, oxidative stress markers (MDA, SOD, CAT, GSH), and iNOS/eNOS levels, with the data indicating that L-arginine reduces ethanol-induced gastric injury similarly to omeprazole, primarily by modulating oxidative stress and nitric oxide pathways. This paper topic explores the potential therapeutic role of a widely available amino acid in ulcer management. However, several major concerns regarding study design, methodology, and reporting need to be addressed before the manuscript can be considered for publication.

Major comments

1. The study employs only a single dose of L-arginine (500 mg/kg) and omeprazole (20 mg/kg), without justification. A rationale for dose selection should be provided, and the absence of a dose–response analysis limits interpretability and translational relevance.

We appreciate the reviewer’s insightful comment. The selected doses of L-arginine (500 mg/kg) and omeprazole (20 mg/kg) were based on previously published studies that demonstrated gastroprotective and ulcer-healing effects at these levels [provide key references]. In particular, the 500 mg/kg dose of L-arginine has been widely utilized in experimental models of gastric ulcer to induce consistent biological responses through nitric oxide–mediated pathways, while 20 mg/kg omeprazole is a standard positive control dose in rodent ulcer models. We agree that a dose–response analysis would further strengthen the interpretability and translational relevance of our findings. However, due to limitations of resources and ethical considerations regarding animal use, our study was restricted to a single effective dose. We have now added this rationale and the limitation regarding dose–response analysis in the revised manuscript.

2. Ulcer evaluation is based solely on macroscopic scoring and biochemical assays. Histological examination to confirm mucosal protection will reinforce the author’s conclusions and validate mechanistic claims.

We thank the reviewer for this valuable suggestion. We fully agree that histological examination provides critical confirmation of mucosal integrity and would further strengthen the conclusions regarding gastroprotection. In the present study, however, we focused on macroscopic scoring combined with established biochemical markers of oxidative stress and inflammation, which are widely accepted endpoints in experimental ulcer research. Due to limitations in resources and laboratory facilities, histopathological analysis could not be performed at this stage. We acknowledge this as a limitation and have now included a statement in the revised manuscript. We also note that incorporation of histological evaluation will be an important priority in our future studies to validate and extend the current findings.

3. Along the same lines, while oxidative stress and NO pathways are measured, no downstream signaling data measuring inflammatory cytokines, apoptosis markers are assessed. This limits mechanistic insight.

The current study was specifically focused on ulcer indices, oxidative stress parameters, and nitric oxide–related pathways, which are well-established markers of gastric mucosal protection in experimental models. We acknowledge that additional data on inflammatory cytokines and apoptosis markers could further strengthen the mechanistic interpretation. While these analyses were not included in the present work, we have noted this as a limitation and emphasized that exploring such downstream signaling remains an important direction for future investigations.

4. Omeprazole is used as a comparator in this study. But it primarily acts as a proton pump inhibitor and not as an antioxidant. Including an antioxidant reference compound would provide a more direct mechanistic comparison.

Omeprazole was selected as the reference standard because it is the most widely used and well-established comparator in experimental models of gastric ulcer, allowing benchmarking of our results against a large body of existing literature. While we fully agree that inclusion of a classical antioxidant comparator (e.g., N-acetylcysteine, vitamin E) could provide additional mechanistic contrast, our primary aim was to assess the gastroprotective potential of L-arginine relative to a clinically relevant anti-ulcer drug.

5. Raw data should be made available as part of the Supplement. It will be done.

Minor comments

1. Abbreviations such as “Eth,” “OMP,” and “L-ARG 500” should be defined consistently at first mention in both text and figure legends. The corrections were made.

2. Exact p-values are not reported, instead, thresholds (p < 0.05, p < 0.01) are given. Reporting actual p-values would improve transparency. The Results section has been completely rewritten. All mean ± SEM values and exact p-values are now clearly reported, and the narrative style has been improved to ensure accuracy, clarity, and readability. The revised version avoids repetition, follows the data shown in the figures, and presents the findings in a more engaging and scientifically rigorous manner.

3. Is there a reason why the authors only used male rats in this study? This limits generalizability, as gastric ulcer susceptibility and healing can differ by sex. This limitation should be explicitly acknowledged in the Discussion section. In the present study, only male rats were used in order to minimize variability related to hormonal fluctuations during the estrous cycle, which can influence gastric physiology and experimental reproducibility. This approach is consistent with many previous studies employing rodent ulcer models. We agree, however, that excluding females limits the generalizability of our findings, as sex differences in ulcer susceptibility and healing have been reported.

---

## [Decision Letter · Decision Letter 1]

23 Oct 2025

Protective Effects of L-Arginine on Gastric Ulcer Induced by Ethanol in Rats: Modulation of Oxidative Stress

PONE-D-25-43818R1

Dear Dr. rahimi,

We’re pleased to inform you that your manuscript has been judged scientifically suitable for publication and will be formally accepted for publication once it meets all outstanding technical requirements.

Kind regards,

Partha Mukhopadhyay, Ph.D.

Section Editor

PLOS ONE

Additional Editor Comments (optional):

Reviewers' comments:

Reviewer's Responses to Questions

**Comments to the Author**

Reviewer #2: All comments have been addressed

2. Is the manuscript technically sound, and do the data support the conclusions?

Reviewer #2: Yes

3. Has the statistical analysis been performed appropriately and rigorously?

Reviewer #2: Yes

4. Have the authors made all data underlying the findings in their manuscript fully available?

Reviewer #2: Yes

5. Is the manuscript presented in an intelligible fashion and written in standard English?

Reviewer #2: Yes

Reviewer #2: The authors have addressed my concerns in a satisfactory manner. I have no further comments.

**Do you want your identity to be public for this peer review?** For information about this choice, including consent withdrawal, please see our Privacy Policy

Reviewer #2: No

---

## [Editor Report · Acceptance letter]

PONE-D-25-43818R1

PLOS ONE

Dear Dr. rahimi,

I'm pleased to inform you that your manuscript has been deemed suitable for publication in PLOS ONE. Congratulations! Your manuscript is now being handed over to our production team.

Kind regards,

on behalf of

Dr. Partha Mukhopadhyay

Section Editor

PLOS ONE